# Mitochondrial Reactive Oxygen Species in TRIF-Dependent Toll-like Receptor 3 Signaling in Bronchial Epithelial Cells against Viral Infection

**DOI:** 10.3390/ijms25010226

**Published:** 2023-12-22

**Authors:** Ga Eul Chu, Jun Young Park, Chan Ho Park, Won Gil Cho

**Affiliations:** 1Department of Anatomy, Yonsei University Wonju College of Medicine, 20 Ilsan-ro, Wonju 26426, Republic of Korea; xiahgaeul@naver.com (G.E.C.); wktlr8@naver.com (C.H.P.); 2Department of Nuclear Medicine, Severance Hospital, Yonsei University College of Medicine, 50-1 Yonsei-ro, Seodaemun-gu, Seoul 03722, Republic of Korea; abies60@naver.com

**Keywords:** mitochondria, reactive oxygen species, TLR3 signaling, bronchial epithelial cells, viral infection

## Abstract

Toll-like receptor 3 (TLR3) plays an important role in double-stranded RNA recognition and triggers the innate immune response by acting as a key receptor against viral infections. Intracellular reactive oxygen species (ROS) are involved in TLR3-induced inflammatory responses during viral infections; however, their relationship with mitochondrial ROS (mtROS) remains largely unknown. In this study, we show that polyinosinic–polycytidylic acid (poly(I:C)), a mimic of viral RNA, induced TLR3-mediated nuclear factor-kappa B (NF-*κ*B) signaling pathway activation and enhanced mtROS generation, leading to inflammatory cytokine production. TLR3-targeted small interfering RNA (siRNA) and Mito-TEMPO inhibited inflammatory cytokine production in poly(I:C)-treated BEAS-2B cells. Poly(I:C) recruited the TLR3 adaptor molecule Toll/IL-1R domain-containing adaptor, inducing IFN (TRIF) and activated NF-*κ*B signaling. Additionally, TLR3-induced mtROS generation suppression and siRNA-mediated TRIF downregulation attenuated mitochondrial antiviral signaling protein (MAVS) degradation. Our findings provide insights into the TLR3-TRIF signaling pathway and MAVS in viral infections, and suggest TLR3-mtROS as a therapeutic target for the treatment of airway inflammatory and viral infectious diseases.

## 1. Introduction

Toll-like receptors (TLRs) are type I transmembrane pattern recognition receptors (PRRs) that play an important role in the innate immune system by recognizing danger-associated molecular patterns (DAMPs) and pathogen-associated molecular patterns (PAMPs) [1]. TLRs act as the first line of host defense against microbial infections, including viruses, bacteria, fungi, and parasites, and contribute to epithelial injury protection and homeostasis maintenance [2,3]. To date, 10 TLRs have been identified in humans (TLR1–TLR10) and 12 have been identified in mice (TLR1–TLR9 and TLR11–TLR13) [4]. TLRs are generally classified into two categories based on their cellular localization. TLR1, TLR2, TLR4–TLR6, and TLR11 are located on the cell surface, whereas TLR3 and TLR7–TLR9 are mainly expressed in intracellular compartments, including the endoplasmic reticulum (ER), endosomes, and lysosomes [5].

Among the TLRs, TLR3 plays a protective role against viral infections. TLR3 specifically recognizes double-stranded RNA and initiates antiviral immune responses via type I interferon (IFN) production and downstream IFN-stimulated gene (ISG) activation [6,7]. All TLRs, except TLR3, recruit myeloid differentiation primary response gene 88 (MyD88) as an adapter protein to induce inflammatory cytokine genes [8]. However, TLR3 is primarily activated via the TIR-domain-containing adapter-inducing interferon-β (TRIF)-dependent pathway to trigger antiviral immune responses [9]. TLR3 can also recognize polyinosinic–polycytidylic acid (poly(I:C)) composed of annealed homopolymers of inosine and cytidine nucleotides [10]. Poly(I:C) is a synthetic dsRNA analog. TLR3 stimulation by poly(I:C) recruits the intracellular adaptor protein TRIF, which activates nuclear factor-kappa B (NF-*κ*B) and IFN regulatory factor 3 (IRF3), which in turn induces proinflammatory cytokine production [11].

Mitochondria are dynamic intracellular organelles responsible for adenosine triphosphate (ATP) generation via oxidative phosphorylation (OXPHOS) [12]. Mitochondria are the main sources of cellular reactive oxygen species (ROS) [13]. During oxidative ATP production, the majority of oxygen (O_2_) molecules are reduced to water in complex IV; however, 1–2% of O_2_ is incompletely reduced to water, leading to ROS generation [14]. ROS are natural byproducts of cellular aerobic metabolism that play an important role in normal physiological activity regulation, immune response mediation, and redox balance maintenance [15,16,17]. TLR signaling increases mitochondrial ROS (mtROS) production in response to invading pathogens, ultimately contributing to the antimicrobial immune defenses [18,19].

Airway epithelial cells are pivotal in innate immunity against airborne pathogens, mainly via TLR activation [20]. TLR3 is constitutively expressed in alveolar and bronchial epithelial cells [21], and directly mediates antiviral host immune responses against dsRNA viruses and viral replication intermediates [22]. TLR3 stimulation triggers intracellular ROS generation in innate immune responses, which contributes to inflammatory cytokine release [23,24]. However, the relationship between TLR3 activation by viral infection and mtROS generation remains largely unexplored.

The current study aims to elucidate the mechanisms underlying dsRNA-induced activation of the TLR3 signaling pathway and subsequent mtROS generation in bronchial epithelial cells. We discovered that poly(I:C)-induced TLR3 stimulation activates the NF-*κ*B signaling pathway, which leads to mtROS generation and proinflammatory cytokine production in the human bronchial epithelial cell line BEAS-2B. The mitochondrial antiviral signaling protein (MAVS) is a crucial adaptor protein in innate immune responses against RNA virus infection [25]. MAVS is activated in retinoic acid-inducible gene I (RIG-I)-like receptor (RLR) signaling pathways, and facilitates TANK-binding kinase-1 (TBK1) activation. TBK1 phosphorylates the transcription factor interferon regulatory factor 3 (IRF3), promoting of IFN transcription [26]. We found that TLR3-dependent mtROS generation influences MAVS degradation. Additionally, siRNA-mediated TRIF knockdown effectively suppressed MAVS degradation and downstream signaling. Our results suggest that TLR3-mtROS signaling is essential in innate antiviral immunity and that TLR3-TRIF signaling negatively regulates MAVS activity.

## 2. Results

### 2.1. Poly(I:C)-Induced TLR3 Activation and NF-κB Signaling Pathway Initiation in Bronchial Epithelial Cells

To assess if human bronchial epithelial cells (BEAS-2B) respond to TLR3 agonists, we examined the impact of poly(I:C) on the expression of phosphorylated TLR3 (p-TLR3) via Western blot analysis. We investigated the time-dependent effects of poly(I:C) on p-TLR3 expression (Figure 1A). In untreated BEAS-2B cells, p-TLR3 expression was notably low. However, poly(I:C) treatment resulted in a significant increase in p-TLR3 protein levels at each examined time point (Figure 1B). Our findings revealed that TLR3 phosphorylation occurred as early as 1 h following poly(I:C) treatment in BEAS-2B.

TLR3 activates inflammatory responses via NF-*κ*B that induces pro-inflammatory cytokine and chemokine synthesis in several cell types [27,28]. Inhibitor kappa B-alpha (I*κ*B-α) phosphorylation is required for NF-*κ*B translocation into the nucleus. To explore the effect of poly(I:C) on NF-*κ*B activation, the phosphorylated I*κ*B-α (p-I*κ*B-α) levels were evaluated in BEAS-2B cells. The results revealed that poly(I:C) treatment significantly increased the p-I*κ*B-α protein levels at 1 h, while I*κ*B-α was rapidly decreased after 1 h poly(I:C) treatment (Figure 1C,D). Subsequently, I*κ*B-α expression was slightly restored from 6 h of treatment. These results suggest that poly(I:C) promotes rapid TLR3 activation and NF-kB pathway induction in BEAS-2B cells.

### 2.2. mtROS Generation Induced in TLR3 Signaling after 6 h of Poly(I:C) Treatment in Bronchial Epithelial Cells

Because mtROS generation can be triggered by various TLR families during inflammatory responses [23,29], we investigated the relationship between TLR3 and mtROS generation in poly(I:C)-treated BEAS-2B cells. To examine whether TLR3 signaling enhances mtROS production, poly(I:C)-stimulated BEAS-2B cells were analyzed via flow cytometry using MitoSOX Red. We first optimized the concentration of MitoSOX for BEAS-2B cells (Appendix A). As shown in Figure 2A, mtROS was gradually generated by poly(I:C) treatment after 6 h of stimulation, and poly(I:C) treatment for 24 h caused a dose-dependent mtROS increase.

The fluorescence intensity of mtROS generation was also observed using confocal microscopy at different concentrations of poly(I:C) at 1 and 24 h (Figure 2B,C). Consistent with the flow cytometry data, poly(I:C) stimulation for 1 h did not increase the level of fluorescence intensity (Figure 2D). However, poly(I:C) significantly increased the mtROS levels after 24 h of treatment (Figure 2E). These results suggested that poly(I:C)-induced TLR3 activation contributes to mtROS generation.

### 2.3. Cytokine Production Affected by Poly(I:C)-Induced mtROS Generation in Bronchial Epithelial Cells

Poly(I:C)-induced ROS generation contributes to inflammatory cytokine production via TLR signaling [24,30,31]. Therefore, we examined whether cytokine production could be modulated by poly(I:C)-induced mtROS generation using the mitochondria-targeted antioxidant Mito-TEMPO. Poly(I:C) stimulated mtROS production; however, it was significantly attenuated by the Mito-TEMPO treatment of BEAS-2B cells (Figure 3A,B). Additionally, we conducted real-time live-cell imaging to confirm mtROS generation following poly(I:C) treatment, with or without Mito-TEMPO (Appendix A). Live cell imaging also demonstrated that the TLR3 agonist induced mtROS generation and that Mito-TEMPO decreased mtROS levels in poly(I:C)-treated BEAS-2B cells.

To confirm that the mtROS stimulates proinflammatory cytokine production, we investigated whether Mito-TEMPO attenuates tumor necrosis factor-alpha (TNF-α) expression using Western blotting (Figure 3C,D). Our results demonstrated that Mito-TEMPO decreased TNF-α protein levels when compared to poly(I:C)-treated BEAS-2B cells. These findings suggest the signaling molecule role whereby mtROS trigger cytokine production in response to dsRNA.

### 2.4. TLR3-Triggered mtROS Contribute to Inflammatory Cytokine Production in Bronchial Epithelial Cells

TLR-mediated ROS generation contributes to inflammatory cytokine production in response to various TLR ligands [32,33]. Therefore, we tested whether blocking TLR3-mediated mtROS generation affects cytokine production in response to poly(I:C) using real-time qPCR analysis (Figure 4A,B). The mRNA expression of TNF-α and IL-6 was significantly increased by poly(I:C) in BEAS-2B cells. However, TLR3-targeted siRNA transfection significantly decreased TNF-α and IL-6 mRNA expression levels in BEAS-2B cells treated with poly(I:C). In addition, Mito-TEMPO significantly decreased TNF-α and IL-6 mRNA levels with or without TLR3 siRNA transfection when compared to poly(I:C)-treated BEAS-2B cells. These findings demonstrate that mtROS are key signaling molecules involved in TLR3-mediated inflammatory responses against dsRNA stimulation.

### 2.5. NF-κB Signaling Pathway Induced by Poly(I:C)-Stimulated TLR3-TRIF Signaling in Bronchial Epithelial Cells

TRIF plays an important role in the TLR3 signaling pathway during poly(I:C) stimulation, and activates NF-*κ*B [34,35]. To verify whether poly(I:C)-induced NF-*κ*B activation relied on the TLR3-TRIF signaling pathway in bronchial epithelial cells, the BEAS-2B cells were transfected with siRNA against TRIF. First, the effect of poly(I:C) on the TRIF and phosphorylated NF-*κ*B (p-NF-*κ*B) p65 expression was evaluated using Western blot analysis. As shown in Figure 5A,B, poly(I:C) significantly increased TRIF and p-NF-*κ*B p65 expression at 6 h. To test the efficiency of TLR3 transcript suppression, BEAS-2B cells were transfected with TLR3-targeted siRNA, and TLR3 and TRIF expression levels were determined. The transfection of BEAS-2B cells with TLR3-targeted siRNA markedly suppressed TLR3 expression upon poly(I:C) stimulation (Figure 5C). In addition, TRIF was not activated by poly(I:C) following TLR3-targeted siRNA treatment, indicating that TRIF is an essential TLR3 adaptor. Next, we examined whether poly(I:C)-induced NF-*κ*B activation depended on the TRIF pathway in BEAS-2B cells (Figure 5D,E). Transfection with TRIF-targeting siRNA effectively suppressed TRIF expression in poly(I:C)-treated BEAS-2B cells. Moreover, TRIF-targeted siRNA specifically attenuated p-I*κ*B-α protein levels. These findings indicate that poly(I:C) induces NF-*κ*B p65 pathway activation via the TLR3-TRIF signaling pathway in bronchial epithelial cells.

### 2.6. MAVS Degradation Affected by Poly(I:C)-Induced mtROS and TLR3-TRIF Signaling in Bronchial Epithelial Cells

The MAVS adaptor protein is crucial in antiviral innate immunity against RNA viral infections [36]. Viral proteins often degrade MAVS as a mechanism to evade the host immune response [37]. To explore the effect of the TLR3 agonist on MAVS expression, we analyzed the protein levels of MAVS and its downstream molecules, TBK1 and IRF3, in BEAS-2B cells using Western blot. As depicted in Figure 6A,B, the MAVS expression initially increased slightly at about 1 h post-treatment with poly(I:C), and then rapidly di-minished at 6 h. Additionally, TBK-1 phosphorylation gradually increased in a time-dependent manner, while IRF-3 was rapidly phosphorylated within 1 h of poly(I:C) treatment. This was followed by a significant decrease in the levels of phosphorylated IRF-3 (p-IRF-3).

To investigate whether poly(I:C)-induced mtROS production affects MAVS degradation, BEAS-2B cells were treated with Mito-TEMPO in the presence or absence of poly(I:C). As shown in Figure 6C,D, Mito-TEMPO significantly restored MAVS protein levels in the presence of poly(I:C). Furthermore, Mito-TEMPO treatment significantly increased the levels of MAVS protein in BEAS-2B cells compared to untreated control cells, indicating that mtROS are closely involved in MAVS protein degradation.

TLR3 and MAVS share common signaling intermediates in antiviral immunity [38,39]. In this study, we demonstrated that the TLR3 agonist, poly(I:C), activates the TRIF-mediated TLR3 signaling pathway in bronchial epithelial cells. Consequently, we hypothesized that MAVS is associated with TRIF when stimulated by TLR3 agonists. To verify this hypothesis, BEAS-2B cells were transfected with TRIF-targeting siRNA, and the expression of MAVS and its downstream molecules was analyzed, both with and without poly(I:C) treatment (Figure 6E,F). We observed that poly(I:C) significantly induced MAVS degradation; however, transfection with TRIF-targeted siRNA specifically attenuated MAVS protein degradation in poly(I:C)-treated BEAS-2B cells. Furthermore, TRIF knockdown also led to the inactivation of TBK-1 and IRF-3. Overall, these findings suggest that the TLR3 adaptor molecule, TRIF, plays a crucial role in modulating MAVS signaling during dsRNA stimulation.

## 3. Discussion

Airway inflammation plays an important role in protecting the host respiratory system against airborne pathogens, particularly viruses. In antiviral immune responses, inflammatory mechanisms use TLR3 to recognize pathogenic molecular patterns. TLR3 activation owing to viral infection triggers type I IFN release from airway epithelial cells to restrict viral replication and activate adaptive immune cell [40,41]. To date, several studies have demonstrated the potential mechanisms of TLR3 in the antiviral response. Intracellular ROS are essential signaling molecules involved in TLR3 signaling [23]. However, the role of mtROS in virus-induced inflammatory signaling and cytokine production in bronchial epithelial cells has not yet been characterized. The results of this study indicated that TLR3 agonists activate the TLR3-mediated NF-*κ*B signaling pathway and trigger mtROS generation, which contributes to inflammatory cytokine production in bronchial epithelial cells. These data further support the role of mtROS in airway inflammation and suggest a potential function of the TLR signaling pathway in viral infections.

ROS are produced in various subcellular compartments, including the plasma membrane, cytoplasm, peroxisomes, mitochondria and endoplasmic reticulum [15]; however, most cellular ROS are generated by mitochondria [42]. Viral infection increases ROS production in immune and infected host cells [43]. For instance, the influenza virus stimulates intracellular ROS overproduction via TLR4 signaling pathway activation, leading to enhanced apoptotic cell death in the lungs [44]. Innate immune cells produce mtROS for use as antimicrobial agents in the host defense against infectious microorganisms [29]. Consistent with previous literature, our study revealed that the poly(I:C), a synthetic analog of viral dsRNA, induces a mtROS increase and TLR3 and I*κ*B-α phosphorylation in human bronchial epithelial cells. mtROS act as signal transducers that release pro-inflammatory cytokines during innate antiviral immunity [45,46]. To further explore whether mtROS contribute to pro-inflammatory cytokine production, we investigated the effect of Mito-TEMPO on cytokine production in poly(I:C)-treated BEAS-2B cells. We confirmed that mitochondria-targeted antioxidant Mito-TEMPO attenuated the levels of mtROS and TNF-α production. dsRNA induces proinflammatory cytokine TLR3 production, which mediates signaling pathways in astrocytes [47]. We further showed that siRNA-mediated TLR3 silencing significantly suppresses mRNA expression and TNF-α and IL-6 production in the presence of poly(I:C). Moreover, the co-treatment of TLR3 siRNA and Mito-TEMPO synergistically suppresses TNF-α and IL-6 expression in poly(I:C)-treated BEAS-2B cells. These results suggest that mtROS are important regulators of virus-induced inflammatory responses.

Upon DAMP and PAMP stimulation, TLRs selectively recruit cytosolic adaptor proteins, including myeloid differentiation primary response gene 88 (MyD88), TIR domain-containing adaptor protein (TIRAP), TRIF-related adaptor molecule (TRAM), sterile α and armadillo motif-containing protein (SARM), B-cell adaptor for phosphoinositide 3-kinase (BCAP), and TRIF, to promote host immune responses [4,48]. TLR signaling pathways are largely divided into two types: MyD88-dependent and TRIF-dependent [49]. TRIF is specifically recruited via TLR3 and TLR4 signaling, activating NF-*κ*B and proinflammatory cytokine production [50]. Consistent with these findings, we confirmed that poly(I:C) upregulates TRIF expression and increases in NF-*κ*B p65 phosphorylation in BEAS-2B cells. Additionally, siRNA-mediated TLR3 silencing suppresses TRIF expression in the presence of poly(I:C). Moreover, TRIF knockdown by siRNA attenuated the phosphorylation of I*κ*B-α in poly(I:C)-treated BEAS-2B cells.

mtROS are generated upon MAVS activation; however, the underlying mechanism remains partially understood [51]. In this study, we demonstrated that Mito-TEMPO treatment reduced MAVS degradation in BEAS-2B cells in response to poly(I:C). Furthermore, to investigate whether the TLR3-dependent signaling pathway contributes to MAVS expression in poly(I:C)-induced antiviral responses, BEAS-2B cells were transfected with TRIF siRNA. We observed that TRIF silencing not only prevented MAVS degradation, but also inactivated its downstream targets such as TBK1 and IRF-3. These findings collectively suggest that MAVS is regulated by mtROS and TLR3-TRIF signaling in bronchial epithelial cells during viral infection.

A significant limitation of our study is its exclusive focus on cellular-level experiments. Further studies should explore whether Mito-TEMPO and siRNAs targeting TLR3 downstream signaling can mitigate proinflammatory cytokine production in a murine model of airway inflammatory disease.

In conclusion, poly(I:C) activated the TLR3-mediated NF-*κ*B signaling pathway and generated mtROS in BEAS-2B cells. Transfection with Mito-TEMPO and TLR3 siRNA synergistically reduced inflammatory cytokine production in poly(I:C)-treated BEAS-2B cells. In addition, Poly(I:C) activates the NF-*κ*B signaling via TLR3-TRIF signaling pathways. Moreover, Mito-TEMPO blocked the poly(I:C)-induced mtROS generation and downregulation of TRIF using siRNA-attenuated MAVS degradation. The TLR3-dependent signaling pathways against viral infections are summarized in Figure 7. These findings suggest an underlying mechanism for the association of mtROS and TLR3-TRIF signaling pathways with MAVS activation in viral infections. Finally, we also identified TLR3-mtROS signaling as a potential therapeutic target for treating airway inflammatory diseases.

## 4. Materials and Methods

### 4.1. Cell Culture and Poly(I:C) Treatment

The BEAS-2B human bronchial epithelial cell line was purchased from the American Type Culture Collection (ATCC, Manassas, VA, USA). BEAS-2B cells were maintained at 37 °C in a humidified 5% CO_2_ atmosphere in a serum-free bronchial epithelial cell growth basal medium (BEGM^®^; Lonza, Walkersville, MD, USA) supplemented with a Bullet Kit (Cambrex Bio Science, Walkersville, MD, USA). The BEGM medium was replaced every second day, and the cells were passaged when they reached 70–80% confluency via incubation with 0.25% trypsin. Poly(I:C) (high molecular weight; Cat# tlrl-pic) was purchased from InvivoGen (San Diego, CA, USA). Poly(I:C) was freshly prepared just prior to each experiment using sterile endotoxin-free water (0.9% NaCl). BEAS-2B cells were seeded onto 100 mm culture dishes at a density of 1 × 10^6^ cells/dish and cultured with a BEGM medium in a humidified CO_2_ incubator at 37 °C. When the cells reached 70–80% confluence, the cells were treated with different concentrations of poly(I:C) (0–100 μg/mL) for up to 24 h.

### 4.2. Transient siRNA Transfections

BEAS-2B cells were seeded at a density of 2 × 10^5^/well in 6-well plates, and cultured until they reached 70–80% confluency. After 18–20 h, cells were transfected with 10 nM (40 pmol) TLR3 siRNA (Cat# sc-36685; Santa Cruz Biotechnology, Dallas, TX, USA) and TRIF siRNA (Cat# s531859; Thermo Fisher Scientific, Waltham, MA, USA) using Lipofectamine 2000 (Cat# 11668-027; Invitrogen, Carlsbad, CA, USA), according to the manufacturer’s protocol. TLR3 or TRIF siRNA and Lipofectamine 2000 (ratio, 1:3) were pre-incubated for 20 min in Opti-MEM before adding them to the seeded cells. The cell medium was replaced with 800 μL fresh Opti-MEM and the prepared mixture was added to each well. At 24 h after siRNA transfection, the Opti-MEM medium was aspirated and replaced with 2 mL complete growth medium containing serum and antibiotics and incubated for 48–72 h. The cells were stimulated with poly(I:C) 10 μg/mL at different time points. Cell lysates were obtained to evaluate siRNA targeting efficiency via immunoblotting.

### 4.3. Western Blot Analysis

Following BEAS-2B cells harvesting using RIPA Buffer, protein samples were quantified using the Pierce BCA Protein Assay Kit (Thermo Fisher Scientific). Protein samples (10 µg) for Western blotting were subjected to electrophoresis on 10% SDS-PAGE gels and transferred onto polyvinylidene fluoride (PVDF) membranes (Bio-Rad Laboratories, Hercules, CA, USA). After 30 min of blocking with 5% BSA at room temperature, membranes were incubated overnight at 4 °C using specific primary antibodies against TLR3 (Cat# ab62566; Abcam, Cambridge, MA, USA) 1:500, phospho-TLR3 (Tyr759) (Cat# PA5-64722, Invitrogen), TRIF (Cat# 4596, Cell Signaling Technology, Danvers, MA, USA) 1:1000, NF-*κ*B p65 (F-6) (Cat# sc-8008, Santa Cruz) 1:2000, phospho-NF-*κ*B p65 (phospho S536) (Cat# ab86299, Abcam) 1:2000, I*κ*B-α (C-21) (Cat# sc-371, Santa Cruz) 1:2000, phospho-I*κ*Bα (Ser32/36) (5A5) (Cat# 9246, Cell Signaling Technology) 1:2000, TNF-α (Cat# 3707, Cell Signaling Technology) 1:1000, and MAVS (Cat# 3993, Cell Signaling) 1:1000 at a specified dilution. Mouse monoclonal anti-GAPDH antibody (Cat# sc-47724, Santa Cruz) 1:5000 was used as the control. The blots were washed and incubated with horseradish peroxidase (HRP)-conjugated secondary antibodies (1:2000 to 1:5000) for 1 h at room temperature. HRP-conjugated goat anti-mouse IgG (Cat# ab205719; Abcam) or HRP-conjugated goat anti-rabbit IgG (Cat# ab97051; Abcam) was used as secondary antibodies. After extensive wash with PBS, protein bands were detected using a Pierce™ ECL Western Blotting Substrate (Thermo Scientific), and blots were scanned using ChemiDoc XRS+ imaging systems (Bio-Rad Laboratories). The intensity of the blot bands was assessed using Image Lab Software (version 6.0.1, Bio-Rad Laboratories).

### 4.4. Flow Cytometry

BEAS-2B cells were seeded at a density of 2 × 10^5^/well in 6-well plates. When the cells reached 70–80% confluence, cells were treated with poly(I:C) (0, 10, or 100 μL/mL) for 1, 3, 6 and 24 h. After washing with Dulbecco’s phosphate-buffered saline (DPBS), cells were stained at 37 °C for 20–30 min with 5 μM MitoSOX (Thermo Fisher Scientific) dissolved in Hank’s balanced salt solution (HBSS). The cells were washed again with DPBS and detached from the plate using a 0.25% trypsin/0.53 mM EDTA solution containing 0.5% polyvinylpyrrolidone (PVP). The cells were then transferred in the FACS tube using growth medium and centrifuged at 200× *g* at 4 °C for 10 min. Samples were resuspended using DPBS and analyzed via flow cytometry on an LSR II Flow Cytometer (Becton Dickinson, Franklin Lakes, NJ, USA), and data were analyzed using FlowJo software (version 10.6.2, Tree Star Software, Ashland, OR, USA).

### 4.5. Confocal Fluorescence Microscopy

The mtROS levels in BEAS-2B cells were measured using MitoSOX (Thermo Fisher Scientific). BEAS-2B cells were seeded on glass coverslips and incubated at 37 °C in 5% CO_2_ for two days. BEAS-2B cells were pretreated with vehicle or 100 μM Mito-TEMPO (Sigma-Aldrich, St. Louis, MO, USA) for 1 h and then stimulated with poly(I:C). After washing with DPBS, cells were stained at 37 °C for 20–30 min with 5 μM MitoSOX dissolved in Hank’s balanced salt solution (HBSS). Then, cells were washed with DPBS, fixed in 4% paraformaldehyde (PFA) for 5 min, and mounted with Vectashield mounting medium containing 4′,6-diamidino-2-phenylindole (DAPI) (Vector Labs, Burlingame, CA, USA). After mounting, the cells were visualized under a confocal microscope (Zeiss LSM-800, Zeiss, Oberkochen, Germany). Imaging analysis was performed using ZEN 2010 image software (version 3.3, Zeiss).

### 4.6. Nano Live Imaging

BEAS-2B cells were seeded at a density of 1 × 10^5^/well in µ-Dish 35 mm Imaging Chamber (4 well, high, ibidi, Gräfelfing Germany). Following pretreatment with or without 100 μM Mito-TEMPO for 1 h, BEAS-2B cells were treated with poly(I:C) and stained with 2 μM MitoSOX. With the μ-Dish located in a 3D Cell Explorer microscope (Nanolive, Ecublens, Switzerland) equipped with a top-stage incubator (Okolab S.R.L., Pozzuoli, Italy), real-time in vitro cell imaging was obtained for 6 h at 37 °C in 5% CO_2_. Images were processed using the Steve software v1.6.3496 (Nanolive).

### 4.7. Real-Time Quantitative PCR

Total BEAS-2B cell RNA was extracted using TRIzol reagent (Invitrogen; Thermo Fisher Scientific). Total RNA (1 µg) was used for complementary DNA synthesis with the ReverTraAce^®^ qPCR RT Master Mix with gDNA Remover (Toyobo, Osaka, Japan), according to the manufacturer’s instructions. Gene expression was analyzed using the SYBR^®^ Green Master Mix (Applied Biosystems, Waltham, MA USA). Specific primers are shown in Table 1. Real-time quantitative PCR (RT-qPCR) was performed under the following amplification conditions: 95 °C for 30 s and 45 cycles of 95 °C for 10 s, 60 °C for 1 min. qPCR products were analyzed in triplicate and run on a real-time PCR system (Applied Biosystems, Foster City, CA, USA). The fold changes of the gene expression were quantified relative to the housekeeping genes, 18S, as an endogenous control, and were quantified using the 2^−ΔΔCT^ method. The Ct values were shown in Appendix A.

### 4.8. Statistical Analysis

All data are presented as the means ± standard deviation (SD). Statistical analyses were performed using the Student *t*-test and a one-way analysis of variance (ANOVA) for multiple comparisons using GraphPad Prism 6 software (San Diego, CA, USA). Differences were considered significant at *p*-values less than 0.05.

## Figures and Tables

**Figure 1 ijms-25-00226-f001:**
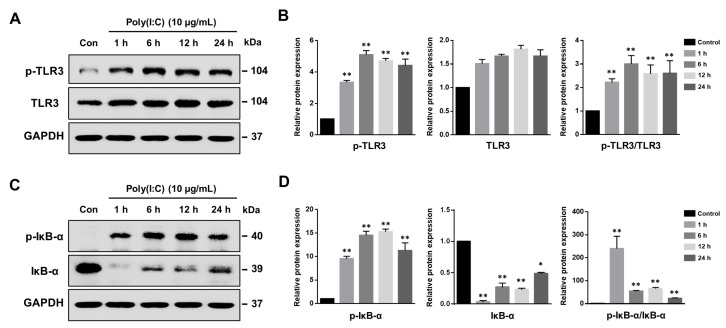
Poly(I:C) induces TLR3 and NF-κB signaling pathway activation. (**A**) BEAS-2B cells were stimulated with 10 μg/mL of poly(I:C) to assess TLR3 phosphorylation. (**B**) Relative protein expressions of phosphorylated TLR3 (p-TLR3), TLR3 and p-TLR3/TLR3 were measured at specific time points and normalized to GAPDH. Data represent the mean ± standard deviation (SD) from three independent experiments, with results normalized to untreated control cells. Statistical significance was assessed using Student’s *t*-test, ** *p* < 0.01 vs. untreated control cells. (**C**) Representative Western blot depicting the protein levels of phosphorylated I*κ*B-α (p-I*κ*B-α) and I*κ*B-α in poly(I:C)-treated BEAS-2B cells over time. (**D**) Quantitative analysis of p-I*κ*B-α, I*κ*B-α and p-I*κ*B-α/I*κB*-α expression levels normalized to GAPDH. Data are the mean ± SD of three independent experiments, and the results normalized to untreated control cells. Statistical analysis was performed via Student’s *t*-test, * *p* < 0.05, ** *p* < 0.01 vs. untreated control cells.

**Figure 2 ijms-25-00226-f002:**
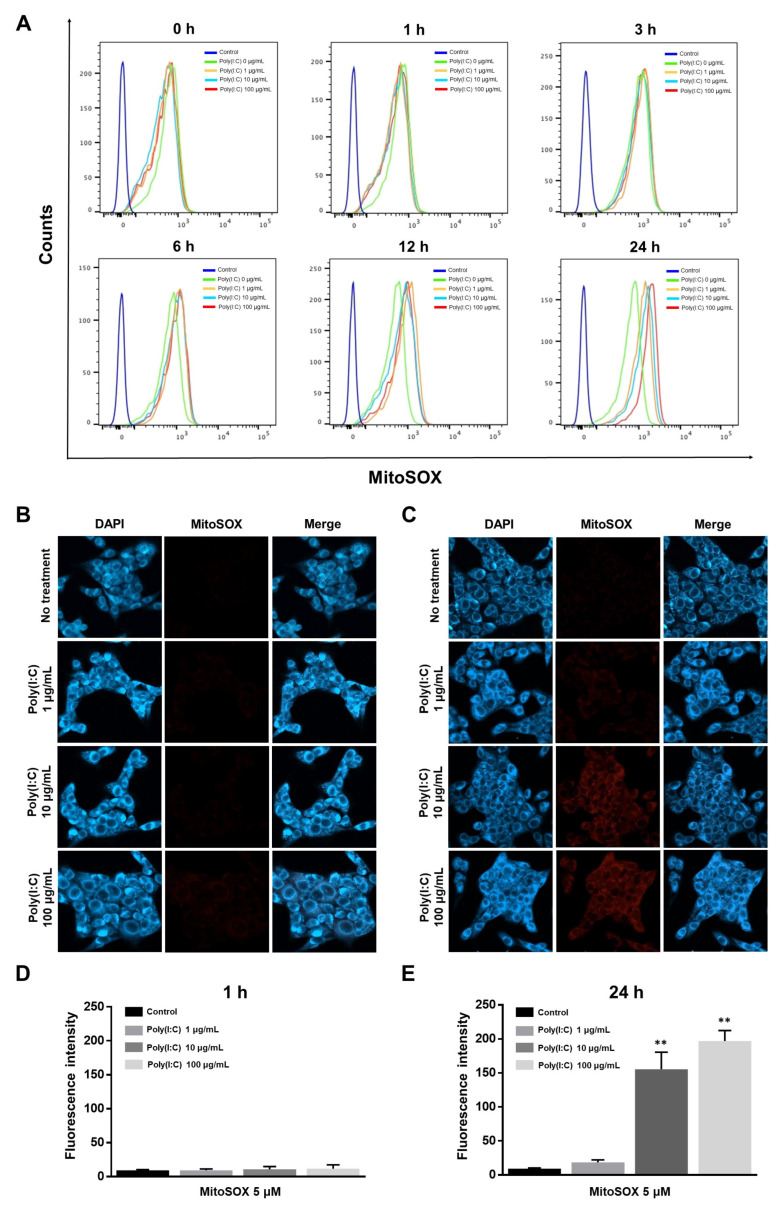
Poly(I:C)-induced mtROS generation. (**A**) BEAS-2B cells were stimulated with poly(I:C) in a time- and dose-dependent manner. mtROS levels were determined via flow cytometry after staining with 5 μM of MitoSOX. Control: non-treated; poly(I:C) 0 μg/mL: treated with only MitoSOX 5 μM. The mtROS levels were assessed via MitoSOX fluorescence at (**B**) 1 h and (**C**) 24 h using confocal microscopy (×400 magnification). BEAS-2B cells were stimulated with poly(I:C) in a dose-dependent manner at different time points, and the mtROS levels were obtained by staining with 5 μM of MitoSOX. (**D**,**E**) Quantitative fluorescence intensity was analyzed using ImageJ software (version 1.53e). Scale bar, 50 μm. Data are the mean ± standard deviation (SD) of three independent experiments, and the results normalized to untreated control cells. Statistical analysis was performed via Student’s *t*-test, ** *p* < 0.01 vs. untreated control cells.

**Figure 3 ijms-25-00226-f003:**
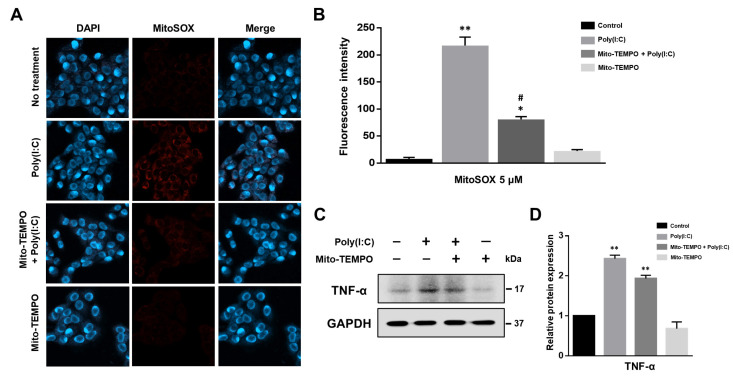
mtROS scavenger Mito-TEMPO attenuates inflammatory cytokine production. (**A**) BEAS-2B cells were pretreated with or without Mito-TEMPO for 1 h, followed by stimulation with 10 μg/mL of poly(I:C). mtROS levels were evaluated using MitoSOX (5 μM) staining, visualized by confocal microscopy at 400× magnification. (**B**) Quantification of MitoSOX fluorescence in BEAS-2B cells across different groups. Fluorescence intensity was quantified using ImageJ software (version 1.53e). Data represent the mean ± standard deviation (SD) from three independent experiments, normalized to untreated control cells. Statistical significance was assessed with Student’s *t*-test, * *p* < 0.05, ** *p* < 0.01 vs. untreated control cells; # *p* < 0.01 vs. poly(I:C)-treated cells. (**C**) Representative Western blot illustrating TNF-α protein levels in poly(I:C)-treated BEAS-2B cells, with or without Mito-TEMPO pretreatment. (**D**) Quantitative analysis of TNF-α expression levels normalized to GAPDH. Data represent the mean ± SD from three independent experiments, normalized to untreated control cells. Statistical analysis was performed using Student’s *t*-test, ** *p* < 0.01 vs. untreated control cells.

**Figure 4 ijms-25-00226-f004:**
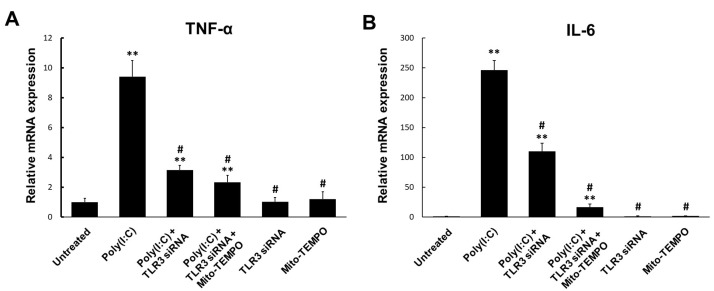
TLR3 silencing reduces inflammatory cytokine induction. BEAS-2B cells were transfected with TLR3 siRNA and treated with 10 μg/mL of poly(I:C) for 6 h with or without Mito-TEMPO pre-treatment. Total BEAS-2B cell RNA was isolated to assess the (**A**) TNF-α and (**B**) IL-6 gene expression using real-time qPCR. Data are the mean ± standard deviation (SD) of three independent experiments, and the results are normalized to untreated control cells. Statistical analysis was performed using Student’s *t*-test, ** *p* < 0.01 vs. untreated control cells; # *p* < 0.01 vs. poly(I:C)-treated cells.

**Figure 5 ijms-25-00226-f005:**
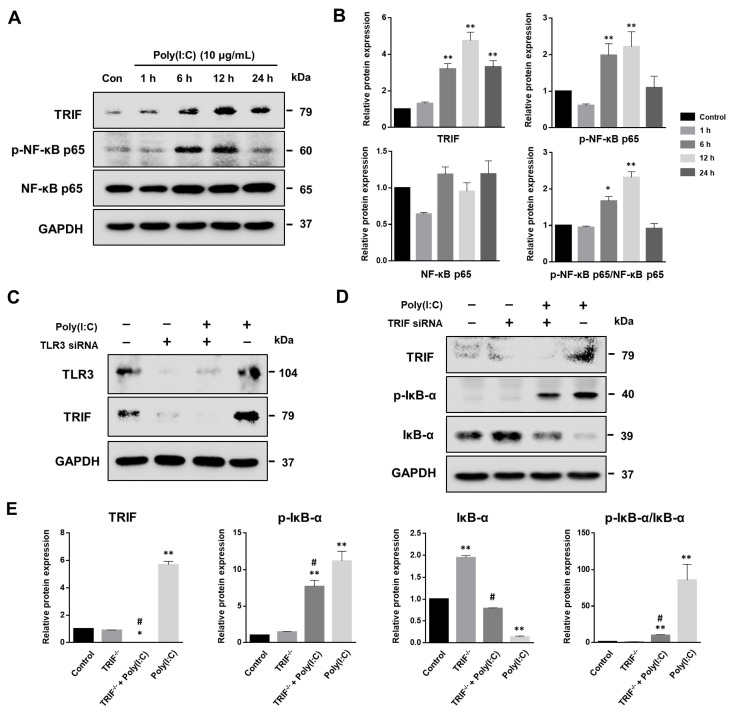
NF-*κ*B pathway activated by TRIF-mediated TLR3 signaling. (**A**) BEAS-2B cells were stimulated with 10 μg/mL poly(I:C) to evaluate TRIF expression and NF-κB p65 phosphorylation. (**B**) Relative TRIF, phosphorylated NF-*κ*B p65 (p-NF-*κ*B p65), NF-*κ*B p65 and p-NF-*κ*B p65/NF-*κ*B p65 protein expressions were assessed according to the indicated time point, and were normalized to the GAPDH reference protein. Data are the mean ± standard deviation (SD) of three independent experiments, and the results normalized to untreated control cells. Statistical analysis was performed using Student’s *t*-test, * *p* < 0.05, ** *p* < 0.01 vs. untreated control cells. (**C**) BEAS-2B cells were pre-treated with 1 μg/mL of TLR3 siRNA and stimulated with 10 μg/mL of poly(I:C) for 6 h. TLR3 and TRIF protein levels detected via Western blot. (**D**) NF-*κ*B signaling molecule expression was evaluated in 1 μg/mL TRIF siRNA-transfected BEAS-2B cells using Western blot analysis. The cells were stimulated with 10 μg/mL of poly(I:C) for 6 h with or without TRIF gene silencing. (**E**) Relative TRIF, phosphorylated I*κ*B-α (p-I*κ*B-α), I*κ*B-α and p-I*κ*B-α/I*κ*B-α protein expressions were assessed and normalized to GAPDH. Data are the mean ± SD of three independent experiments, and the results normalized to untreated control cells. Statistical analysis was performed using Student’s *t*-test, * *p* < 0.05, ** *p* < 0.01 vs. untreated control cells; # *p* < 0.01 vs. poly(I:C)-treated cells.

**Figure 6 ijms-25-00226-f006:**
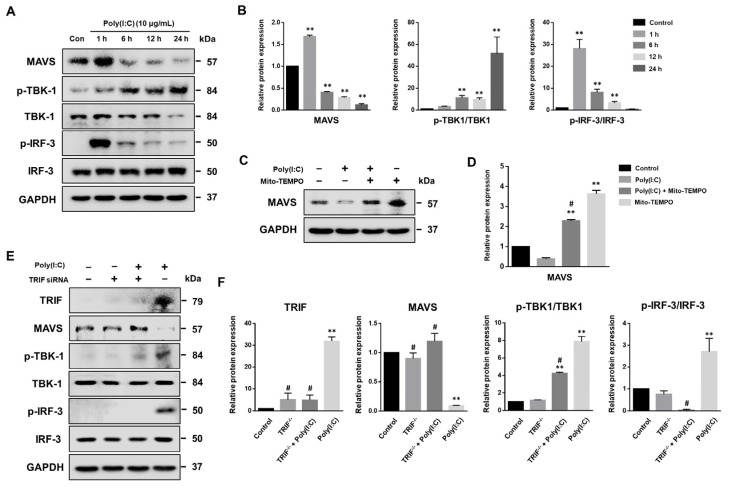
MAVS degradation is inhibited by Mito-TEMPO and TRIF siRNA. (**A**) BEAS-2B cells were stimulated with 10 μg/mL of poly(I:C) and the expression of MAVS signaling pathway analyzed using Western blot. (**B**) Relative MAVS, phosphorylated TBK-1 (p-TBK-1)/TBK-1 and p-IRF-3/IRF-3 protein expressions were assessed according to the indicated time point and normalized to the GAPDH reference protein. Data are the mean ± standard deviation (SD) of three independent experiments, and the results normalized to untreated control cells. Statistical analysis was performed using Student’s *t*-test, ** *p* < 0.01 vs. untreated control cells. (**C**) BEAS-2B cells were pre-treated with 100 μM Mito-TEMPO for 1 h and stimulated with 10 μg/mL of poly(I:C) for 6 h. Expressed MAVS protein levels were detected via Western blot. (**D**) Relative MAVS protein expressions were assessed according to the indicated time point and normalized to the GAPDH reference protein. Data are the mean ± SD of three independent experiments, and the results normalized to untreated control cells. Statistical analysis was performed using Student’s *t*-test, ** *p* < 0.01 vs. untreated control cells; # *p* < 0.01 vs. poly(I:C)-treated cells. (**E**) The expression of MAVS and its downstream molecules were evaluated in TRIF siRNA transfected BEAS-2B cells using Western blot analysis. The poly(I:C)-stimulated cells (6 h) with or without TRIF gene silencing. (**F**) Relative TRIF, MAVS, p-TBK-1/TBK-1 and p-IRF-3/IRF-3 protein expressions were assessed and normalized to GAPDH. Data are the mean ± SD of three independent experiments, and the results normalized to untreated control cells. Statistical analysis was performed using Student’s *t*-test, ** *p* < 0.01 vs. untreated control cells; # *p* < 0.01 vs. poly(I:C)-treated cells.

**Figure 7 ijms-25-00226-f007:**
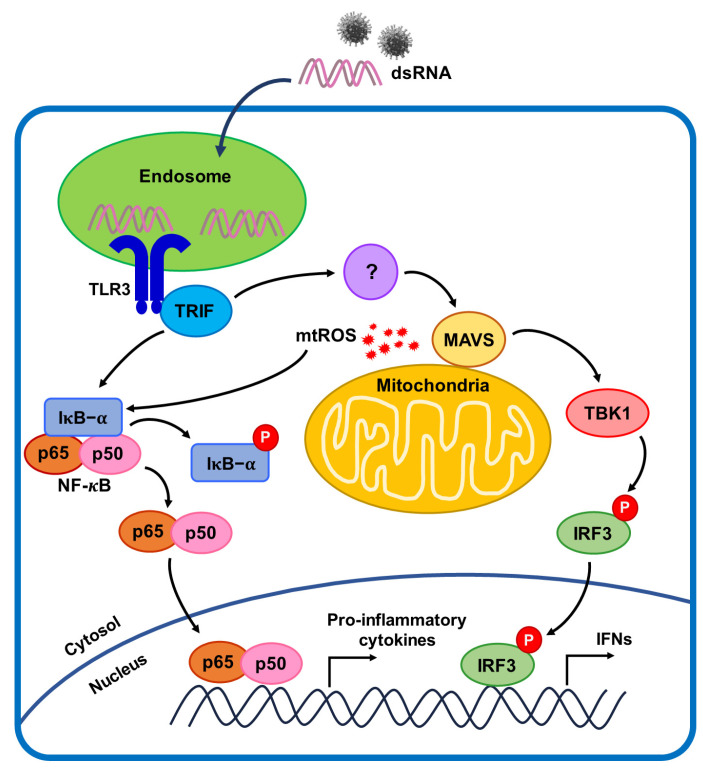
TLR3-dependent NF-*κ*B signaling mechanism associated with MAVS.

**Table 1 ijms-25-00226-t001:** List of RT-qPCR primers used.

Gene	Primer Sequence
IL-6	forward primer: 5′-GGT ACT CCT CCA CGG CAT CT-3′ reverse primer: 5′-GTG CCT CTT TGC TGC TTT CAC-3′
TNF-α	forward primer: 5′-GCC CAT GTT GTA GCA AAC CC-3′ reverse primer: 5′-TAT CTC TCA GCT CCA CGC CA-3′
18S	forward primer: 5′-ACC GCA GCT AGG AAT AAT GGA-3′ reverse primer: 5′-CGG TCA GTT CCG AAA ACC A-3′

## Data Availability

The data presented in this study are available on request from the corresponding author.

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
