# Peer review of "Mitochondrial Reactive Oxygen Species in TRIF-Dependent Toll-like Receptor 3 Signaling in Bronchial Epithelial Cells against Viral Infection"

_ijms, 2023, doi:10.3390/ijms25010226_

Round 1
Reviewer 1 Report
Comments and Suggestions for Authors
The article is devoted to the study of the TLR3-TRIF pathway of the immune response to a viral infection. The text of the article requires revision; after addressing the comments and answering the questions, the article can be published.
The comments include requirements and recommendations.
1) ‘poly(I:C)’ (line 17), ‘TNF-α’ (line 152): Decipher it.
2) ‘…in BEAS-2B cells…’ (lines 74-75): Explain what kind of cells these are.
3) Introduction (line 30)
‘Finally, briefly mention the main aim of the work and highlight the main conclusions.’ (https://www.mdpi.com/journal/ijms/instructions): You emphasized the main conclusions (probably you should have done this more generally, in general terms), but did not discuss the purpose of the study. Introduction needs a purpose.
4) ‘…p-TLR3 expression was absent in untreated BEAS-2B 86 cells’ (lines 86-87): Figure 1A contradicts the thesis. If there is a band, then there is still expression. Reformulate the sentence.
5) ‘however, poly(I:C) induced an increase in p-TLR3 protein expression in a time-dependent manner’ (lines 87-88): The figure shows the difference between control (-) and treatment (+), but the difference between ‘+’-1h and ‘-‘-24h is not obvious. And the statement sounds as if these two time points differ markedly in expression. Reframe.
6) 2.1. Poly(I:C)-induced TLR3 activation and NF-κB signaling pathway initiation in bronchial epi-82 thelial cells (line 82)
Have you compared the 1st hour with the 6th? 6th with 24th (for prediction phosphorylation dynamics, for example)? (Figure 1C)
Was there really no significant difference between the control and 1 and 24 hours? You could discuss this. (Figure 1E)
7) Figure 2 (line 129)
Figure 1A: Add axis labels.
‘Control; non-treated, Poly(I:C) 0 μg/mL; treated only MitoSOX 5 μM’ (line 131): Perhaps you meant this:‘Control: non-treated; Poly(I:C) 0 μg/mL: treated only MitoSOX 5 μM’. Reword it so it's clear.
8) ‘Poly(I:C) stimulated mtROS production; however, it was significantly attenuated by Mito-TEMPO treatment of BEAS-2B cells’ (lines 145-146): This conclusion can be drawn if we compare the signals during infection with and without the antioxidant (Poly(I:C) vs. Poly(I:C)+Mito-TEMPO). Have you made such a comparison? If yes, then this should be indicated in Figure 3B.
9) Figure S1 (line 148): It would be better if the sequence of videos in Supplementary matches the order in the article: no-treatment -> Poly(I:C) -> Mito-TEMPO+ Poly(I:C) -> Mito-TEMPO. It will be clearer this way.
10) 2.5. NF-κB signaling pathway induced by poly(I:C)-stimulated TLR3-TRIF signaling in bronchial epithelial cells (line 189): There are possible interesting comparisons that are not discussed, such as the decrease in p-NF-κB protein after 1 hour. Why isn't this being discussed?
11) ‘Furthermore, Mito-TEMPO treatment significantly increased the levels of MAVS protein in BEAS-2B cells compared to untreated control cells’ (lines 237-238): Comparison data are not provided.
12) Figure 6 (line 253): Figures 6A and Figures 6B look contradictory (according to Figures 6A, MAVS increases after an 1 hour compared to the control, and according to Figures 6B it decreases after an 1 hour compared to the control). How do you explain this?
13) Figure 7 (line 262): The figure should be where you refer to it (line 329).
14) ‘…medium containing serum and antibiotics…’ (line 360): Do you think that the use of an antibiotic on cell culture could have distorted the analysis?
15) ‘The gene-specific primers used were as follows: IL-1β, 5’-TTA AAG CCC GCC TGA CAG A and 5’-GCG 424 AAT GAC AGA GGG TTT CTT AG; IL-6, 5’-GGT ACT CCT CCA CGG CAT CT and 5’-GTG CCT CTT TGC TGC TTT CAC; TNF-α, 5’-GCC CAT GTT GTA GCA AAC CC and 5’-TAT CTC TCA GCT CCA CGC CA; and 18S, 5’-ACC GCA GCT AGG AAT AAT GGA and 5’-CGG TCA GTT CCG AAA ACC A’ (lines 423-427): It is better to present this in the form of a table; it is difficult to perceive in the text.
16) ‘The fold changes of the gene expression were quantified relative to the housekeeping genes’ (lines 431-432): List the genes you normalized to.
17) As for original images for blots, it was a bit hard to evaluate the information without molecular weight of the target proteins.
Author Response
Point-by-point response to Comments and Suggestions for Authors
Comments 1: ‘poly(I:C)’ (line 17), ‘TNF-α’ (line 152): Decipher it
Response 1: Thank you for your detailed review. We added information according to your suggestion. (line 17, 161)
Comments 2: ‘…in BEAS-2B cells…’ (lines 74-75): Explain what kind of cells these are.
Response 2: Thank you for your detailed review. We added information of the BEAS-2B cells according to your suggestion. (line 72)
Comments 3: Introduction (line 30)
‘Finally, briefly mention the main aim of the work and highlight the main conclusions.’ (https://www.mdpi.com/journal/ijms/instructions): You emphasized the main conclusions (probably you should have done this more generally, in general terms), but did not discuss the purpose of the study. Introduction needs a purpose.
Response 3: Thank you for your suggestion. In accordance with your comments, we added the purpose of the study in the introduction of revised version. (lines 71-73)
Comments 4: ‘…p-TLR3 expression was absent in untreated BEAS-2B 86 cells’ (lines 86-87): Figure 1A contradicts the thesis. If there is a band, then there is still expression. Reformulate the sentence.
Response 4: Thank you for your comment. As per your suggestion, the sentence has been changed to “In untreated BEAS-2B cells, p-TLR3 expression was notably low.” in the revised manuscript. (lines 91-92)
Comments 5: ‘however, poly(I:C) induced an increase in p-TLR3 protein expression in a time-dependent manner’ (lines 87-88): The figure shows the difference between control (-) and treatment (+), but the difference between ‘+’-1h and ‘-‘-24h is not obvious. And the statement sounds as if these two time points differ markedly in expression. Reframe.
Response 5: Thank you for your detailed review. We agree with your opinion. The sentence has been changed to “However, poly(I:C) treatment resulted in a significant increase in p-TLR3 protein levels at each examined time point.” in the revised manuscript. (lines 92-93)
Comments 6: 2.1. Poly(I:C)-induced TLR3 activation and NF-κB signaling pathway initiation in bronchial epi-82 thelial cells (line 82)
Response 6: Thank you for your detailed review.
Have you compared the 1st hour with the 6th? 6th with 24th (for prediction phosphorylation dynamics, for example)? (Figure 1C)
As shown Figure 1B, TLR3 was phosphorylated in 1 h of poly(I:C) treatment and increased by 6 h. And then, the expression of p-TLR3 was slightly decreased from 12 h treatment. We added p-TLR3/TLR3 relative protein expression to compare the prediction phosphorylation.
Was there really no significant difference between the control and 1 and 24 hours? You could discuss this. (Figure 1E)
We added the p-IκB-α/ IκB-α relative protein expression. As shown figure E, the expression of p-IκB-α significantly differed between the control and 1 and 24 hours. We also mentioned that IκB-α was rapidly decreased within 1 h of poly(I:C) treatment, followed by a slight resynthesis in revised version. (lines 100-102)
Comments 7: Figure 2 (line 129)
Figure 1A: Add axis labels.
‘Control; non-treated, Poly(I:C) 0 μg/mL; treated only MitoSOX 5 μM’ (line 131): Perhaps you meant this:‘Control: non-treated; Poly(I:C) 0 μg/mL: treated only MitoSOX 5 μM’. Reword it so it's clear.
Response 7: Thank you for your suggestion. As per your suggestion, we added the axis labels in Figure 2A. The sentence was revised as you mentioned in revised version. (line 140)
Comments 8: ‘Poly(I:C) stimulated mtROS production; however, it was significantly attenuated by Mito-TEMPO treatment of BEAS-2B cells’ (lines 145-146): This conclusion can be drawn if we compare the signals during infection with and without the antioxidant (Poly(I:C) vs. Poly(I:C)+Mito-TEMPO). Have you made such a comparison? If yes, then this should be indicated in Figure 3B.
Response 8: Thank you for your comment. The comparison between poly(I:C) and poly(I:C)+Mito-TEMPO was analyzed and added in the legend of Figure 3B in the revised version. (line 176)
Comments 9: Figure S1 (line 148): It would be better if the sequence of videos in Supplementary matches the order in the article: no-treatment -> Poly(I:C) -> Mito-TEMPO+ Poly(I:C) -> Mito-TEMPO. It will be clearer this way.
Response 9: Thank you for your detailed review. As per your suggestion, we rearranged the sequence of videos in the revised version of the Supplement (Figure S2).
Comments 10: 2.5. NF-κB signaling pathway induced by poly(I:C)-stimulated TLR3-TRIF signaling in bronchial epithelial cells (line 189): There are possible interesting comparisons that are not discussed, such as the decrease in p-NF-κB protein after 1 hour. Why isn't this being discussed?
Response 10: Thank you for your detailed review. As per your comment, we performed additional experiment and replaced the data in revised version. The expression levels of p-NF-κB appear to decrease after 1 h, but there was no significant difference in relative protein expression of p-NF-κB/NF-κB (Figure 5B).
Comments 11: ‘Furthermore, Mito-TEMPO treatment significantly increased the levels of MAVS protein in BEAS-2B cells compared to untreated control cells’ (lines 237-238): Comparison data are not provided.
Response 11: Thank you for your comment. As per your suggestion, we added comparison data in Figure 6D.
Comments 12: Figure 6 (line 253): Figures 6A and Figures 6B look contradictory (according to Figures 6A, MAVS increases after an 1 hour compared to the control, and according to Figures 6B it decreases after an 1 hour compared to the control). How do you explain this?
Response 12: Thank you for your detailed review. We apologize for the confusion about the treatment time. In the experiment of Figure 6C, we treated the poly(I:C) for 6 h. Based on your comment, we added the treatment time of in legend of Figure 6 of revised version. (line 285)
Comments 13: Figure 7 (line 262): The figure should be where you refer to it (line 329).
Response 13: Thank you for your suggestion. As per your suggestion, we mentioned the Figure 7 in discussion section. (line 363)
Comments 14: ‘…medium containing serum and antibiotics…’ (line 360): Do you think that the use of an antibiotic on cell culture could have distorted the analysis?
Response 14: Thank you for your comment. Although distorted effect depends on the cells or reaction conditions, Liposome can rapidly decrease their efficacy in responding to the antibiotics, therefore, it is effective to use antibiotics free medium when dealing with cells with low transfection efficacy. In addition, lipofectamine used for transfection can increase the toxicity of responding to the antibiotics, so the condition-sensitive cells are not recommended to add antibiotics.
Comments 15: ‘The gene-specific primers used were as follows: IL-1β, 5’-TTA AAG CCC GCC TGA CAG A and 5’-GCG 424 AAT GAC AGA GGG TTT CTT AG; IL-6, 5’-GGT ACT CCT CCA CGG CAT CT and 5’-GTG CCT CTT TGC TGC TTT CAC; TNF-α, 5’-GCC CAT GTT GTA GCA AAC CC and 5’-TAT CTC TCA GCT CCA CGC CA; and 18S, 5’-ACC GCA GCT AGG AAT AAT GGA and 5’-CGG TCA GTT CCG AAA ACC A’ (lines 423-427): It is better to present this in the form of a table; it is difficult to perceive in the text.
Response 15: Thank you for your detailed review. As per your suggestion, gene-specific primers are listed in the Table 1. (lines 470, 471)
Comments 16: ‘The fold changes of the gene expression were quantified relative to the housekeeping genes’ (lines 431-432): List the genes you normalized to.
Response 16: Thank you for your comment. We mentioned housekeeping gene, 18s in Material & method 4.7 Real Time Quantitative PCR section and listed the gene sequence in table 1. (lines 463, 468)
Comments 17: As for original images for blots, it was a bit hard to evaluate the information without molecular weight of the target proteins.
Response 17: Thank you for your detailed review. We agreed with your comment. As for your suggestion, we added the molecular weight of each proteins in revised manuscript. (original images for blots as well)

Reviewer 2 Report
Comments and Suggestions for Authors
The presented paper describes the role of TLR3-NfKB axis under mimetic response to Poly(I:C).
The presented area has multiple open questions in terms of innate immune responses, so I went over the paper with a great interest. After carefully review, I suggest performing additional experiment and then resubmit the paper.
Please follow my majors that preclude publication in its current form.:
- Used cell line (BAES-2B) is typical epithelial cell line that is primarly used to study respiratory ion transport as well as the function of CFTR. For viral infection more dynamic cell lines should be used, like freshly isolated neutrophils that are the first line of the antiviral response.
- The shown phosphorylation of TLR3 as a response to dsRNA stimuli has been in-depth described 10 years ago -10.1089/jir.2014.0034
- The shown mitoSox generation by TLR3/NfKB dependent manner has been already shown and well studied - 10.1007/s00018-019-03148-8
- The suggestion of the involving MAVS in the observed phenomena requires further verification - whether the MAVS is involved or not. MAVS usually is the key response for RiG-I dependent reply to ssRNA via TRIM25 K63specific ubiquitination. Later on, it leads to IRF3 phosphorylation through the TBK1/IKKepsilon related pathway and the type-I IFN response. The way linking MAVS, dsRNA, TLR3 and NfKB must be evaluated by excluding proposed pathway and confirmation of the activation of signal transducer from TLR3 to NfKB by assesing: IKK alpha/betta/epsilon function. Now, there is no proof for the existence of the proposed link.
- the result suggests that after 6 post stimulation in the used cell line the levels of TRIF are significantly higher than after 1 hour. It is truly questionable since this cell line is considered as a slow response cell line and the TRIF machinery has an adaptory nature, thus in isolated environment (cell line culture) this response is absent or very slow. Usually epithelial cells are stimulated by cargo (eg EVs) released by neutrophils, monocytes or macrophages.
- using a Poly(IC) mimic has been already linked multiple time for TLR family activity. With regret, I must admit that using only cell line, mimic agent and basic techniques as westrns and mitosox makes this paper wit very low novelty and questionable results.
Author Response
Point-by-point response to Comments and Suggestions for Authors
Comments 1: Used cell line (BAES-2B) is typical epithelial cell line that is primarily used to study respiratory ion transport as well as the function of CFTR. For viral infection more dynamic cell lines should be used, like freshly isolated neutrophils that are the first line of the antiviral response.
Response 1: Thank you for your comment. BEAS-2B cell line is bronchial epithelial cells widely used to study the inflammation and various related signaling pathways. Since epithelial cells play an important role in the innate immune responses to virus infection, we used BEAS-2B cell line in this study. As per your suggestion, we were able to obtain the better result with immune cells such as neutrophils. In further study, we will use a more dynamic cell line as you mentioned and investigate the responses to immune cells. We thank for your valuable feedback.
Comments 2: The shown phosphorylation of TLR3 as a response to dsRNA stimuli has been in-depth described 10 years ago -10.1089/jir.2014.0034
Response 2: Thank you for your comment. We identified expression levels of p-TLR3 in order to obtain fundamental data to elucidate a novel mechanism between MAVS and TRIF in the present experiment. Based on your comment, we have added the suggested article in the revised manuscript of the Reference [6].
Comments 3: The shown mitoSox generation by TLR3/NfKB dependent manner has been already shown and well-studied - 10.1007/s00018-019-03148-8
Response 3: Thank you for suggesting this reference. We have carefully reviewed article 10.1007/s00018-019-03148-8 (Proinflammatory NF-kB signalling promotes mitochondrial dysfunction in skeletal muscle in response to cellular fuel overloading). However, there was a lack of evidence that mtROS caused by TLR3 expression in the suggested article. In this study, we tried to demonstrate the relationship between TLR3 and mtROS using mitoSOX.
Comments 4: The suggestion of the involving MAVS in the observed phenomena requires further verification - whether the MAVS is involved or not. MAVS usually is the key response for RiG-I dependent reply to ssRNA via TRIM25 K63specific ubiquitination. Later on, it leads to IRF3 phosphorylation through the TBK1/IKKepsilon related pathway and the type-I IFN response. The way linking MAVS, dsRNA, TLR3 and NfKB must be evaluated by excluding proposed pathway and confirmation of the activation of signal transducer from TLR3 to NfKB by assesing: IKK alpha/betta/epsilon function. Now, there is no proof for the existence of the proposed link.
Response 4: Thank you for your detailed review. We agreed with your comment. Based on your comment, we performed additional experiment to support evidence for the mechanism. TBK 1 and IRF-3 are normally phosphorylated with poly(I:C) treatment, however, TRIF knockdown inhibited expression of p-TBK1 and IRF-3. We added the new figure and mentioned corresponding results in revised version. (lines 268, 271-272)
We would like to thank the reviewer for this comment.
Comments 5: the result suggests that after 6 post stimulation in the used cell line the levels of TRIF are significantly higher than after 1 hour. It is truly questionable since this cell line is considered as a slow response cell line and the TRIF machinery has an adaptory nature, thus in isolated environment (cell line culture) this response is absent or very slow. Usually epithelial cells are stimulated by cargo (eg EVs) released by neutrophils, monocytes or macrophages.
Response 5: Thank you for your comment. We found that TRIF expression was increased at 1 hour after TLR3 activation by poly(I:C) treatment (Figure 5A). As you mentioned, epithelial cells may be slow to response to stimulants. TRIF is known as a direct adaptor protein of TLR3. TLR3 was significantly activated at 1 h following poly I:C stimulation. Therefore, it is thought that TRIF, a sub-signaling material of TLR3, was also rapidly activated.
Comments 6: using a Poly (IC) mimic has been already linked multiple time for TLR family activity. With regret, I must admit that using only cell line, mimic agent and basic techniques as westrns and mitosox makes this paper with very low novelty and questionable results.
Response 6: Thank you for your valuable comment. This study aimed to elucidate the association between virus-sensitive MAVS and TRIF, a downstream molecule of TLR3, which is activated by dsRNA stimulation. MAVS is known to be activated by RIG-1, but our results demonstrated that MAVS signaling is also associated with TRIF when stimulated by dsRNA.

Reviewer 3 Report
Comments and Suggestions for Authors
Please see attachment.

The quality of English language is overall good, some points are made in the attachment.
Author Response
Point-by-point response to Comments and Suggestions for Authors
Comments 1: Major comment throughout the manuscript: the family of NF-kB transcription factors consists of 5 members and is activated by 2 major pathways. The authors constantly refer to NF-kB in this manuscript while they basically mean the p65/RelA subunit and the canonical pathway. Therefore, they have to be specific and refer to p65/NF-kB or phospho-p65/NFkB wherever it is mentioned.
Response 1: Thank you for your detailed review. We agree with you comment. As per your suggestion, we added p65 to the corresponding word in the revised version.
Comments 2: L64-80: This last paragraph is confusing. The authors mix references for immune cells with comments about epithelial cells. It requires rewriting and clear statements about the immune cells (eg macrophages) and the epithelial cells, such as the cell line that was used in this study.
Response 2: Thanks for this precious advice. In accordance with the reviewers’ comments, we have changed ‘innate immune cells’ to ‘innate immune responses’ in the Introduction of the revised manuscript. We mentioned the previously published journal that epithelial cells play an important role in the innate immune responses to virus infection in Reference [20]. (line 68)
Comments 3: L75-76: The sentence about MAVS suddenly appears at the end of the last paragraph. It must go earlier and provide enough information about its function in immune cells and epithelial cells, enough for the reader to understand the flow of the text.
Response 3: Thank you for your comment. As per your suggestion, we added the information about MAVS and its downstream signaling in the Introduction section of the revised version. (lines 75-80)
Comments 4: Figure 1: the experiments in Fig1A were done with 100ug.ml PolyI:C, but in Fig 1B with 10ug/ml.
• Why different amounts were used? We need to see a dose-effect graph to understand.
Response 4: We apologize for our mistake. In Figure 1A, we first conducted an experiment to determine the effect of TLR3 agonist using 10μg/mL of poly(I:C) at different time points. Since Figure 1A and 1C are the same experiment, we deleted the Figure A in the revised version.
Comments 5: We need an additional graph to 1C that has the normalised values, pTLR3/TLR3
Response 5: Thank you for your comment. As per your suggestion, we added the normalized graph of pTLR3/TLR3 in the revised version. (Figure 1B)
Comments 6: Fig1C: in general, upon activation of the NFkB pathway the IkBa is degraded and then resynthesized, because IkBa is an NF-kB target. Also, the same experiment in the same cell line was published by Bach NS, Låg M, Øvrevik J. Toll like receptor-3 priming alters diesel exhaust particle-induced cytokine responses in human bronchial epithelial cells. Toxicol Lett. 2014 Jul 3;228(1):42-7. doi: 10.1016/j.toxlet.2014.03.021. and they showed slight degradation, and they could see resynthesis at 4h post stimulation.
• Why there is no IkBa resynthesis upon TLR3 activation and how these data can be explained.
Response 6: Thank you for your detailed referenced. We agree with your opinion. Based on your comment, we performed the experiment again, and we were able to obtain data similar to the reference you suggested. We found the resynthesis of IκB-α and added the results in Figure 1C.
Comments 7: The authors mention that all reagents were done in DMSO. What is the final concentration of the DMSO and what is the effect of DMSO on these experiments? The authors need to show data with DMSO alone.
Response 7: We apologize for our mistake. To store the poly(I:C) for a long time, poly(I:C) must be dissolved in DMSO and stored frozen. However, in this study, in order to use fresh poly(I:C), we freshly prepared the poly(I:C) with sterile endotoxin-free physiological water (0.9% NaCl) provided by the manufacturer (Cat# tlrl-pic; InvivoGen, San Diego, CA, USA) just prior to each experiment. Sorry for the confusion. We have modified Material & Method of 4.1. Cell culture and Poly I:C treatment in the revised version. (lines 379-381)
Comments 8: Figure 2: Fig2A, time 0h.
• What the data in Fig2A 0h represent?
Response 8: Thank you for your comment. 0 h in Fig. 2A indicate that the treatment with only mitoSOX. Based on your comment, The sentence ‘Control; non-treated, Poly(I:C) 0 μg/mL; treated only MitoSOX 5 μM’ has been changed to ‘Control: non-treated; Poly(I:C) 0 μg/mL: treated only MitoSOX 5 μM’. Thank you for your suggestion.
Comments 9: Why the amount of ROS at time 0 is very high? Which is contrast to similar time point in Fig2B (confocal data)
Response 9: We measured the amount of ROS at the different concentration of poly(I:C) in Fig. 2B. The amount of ROS at time 0 was detected by FACS in Figure 2A and the shift was basic fluorescence expression of mitoSOX in BEAS-2B cells. We optimized the final concentration of MitoSOX and added this data in Supplemental Figure 1 in the revised version.
Comments 10: Figure 3C: TNF is synthesised as pre-TNF which is cleaved to mature TNF. The antibody used in this study can detect both forms. Therefore, the authors need to present the levels of both pre-TNF and mature TNF in western blot. This will show whether treatment the synthesis or the processing to mature or both.
Response 10: Thank you for your detailed review. We agree with your comment. Based on your comment, we indicated the molecular weight in the Figure 3C. Additionally, the original image for blot we attached shows the levels of both pre-TNF-α and mature TNF-α at each molecular weight.
Comments 11: The western blot is not convincing that the Miso-TEMPO reduces poly I:C-induced TNF levels. There is no quantification / statistics.
Response 11: Thank you for your comment. As per your suggestion, we added the quantification data in Figure 3D.
Comments 12: Figure 4: L177-178: the statement is wrong. The data in Figure 4 show the role of TLR3 and not that mtROS are key signalling molecules involved in TLR3-mediated innate immune responses. ALSO, these responses are not immune response because all the experiments are done on epithelial cells. Therefore, the outcomes should refer to epithelial cells and not immune cells.
Response 12: Thank you for your detailed review. as per your suggestion, the confusing word ‘innate immune responses’ was modified to ‘inflammatory responses’ in the introduction section of the revised manuscript. (line 194)
Furthermore, the sentence “In addition, Mito-TEMPO significantly decreased TNF-α and IL-6 mRNA levels with or without TLR3 siRNA transfection when compared to poly(I:C)-treated BEAS-2B cells.” has been added in the revised version. (lines 191-193)
Comments 13: • Figure 4 experiments are missing essential controls, eg TLR3siRNA alone and Mito-TEMPO alone and solvent alone
Response 13: Thank you for your detailed review. We agreed with your comment. As per your suggestion, we conducted the experiment again and added new data in the revised manuscript. (Figure 4)
Comments 14: Consequently the analysis DDCt method is wrong because the reference sample is not the untreated for all conditions. The untreated can be used as reference only for the Poly I:C treatment.
Response 14: Thank you for your comment. As you mentioned, we added the untreated for all conditions (TLR3 siRNA alone and Mito-TEMPO alone) and obtained the relative expression data. We added new data in the revised manuscript. (Figure 4)
Comments 15: The figure legend says that the experiment was done with 10ug/ml Poly I:C, but in materials and methods says 1 ug/ml. The authors have to be consistent about experimental procedures.
Response 15: Thank you for your detailed review. we apologize our mistake. As per your suggestion, the concentration shown in the Materials and methods 4.2 Transient siRNA transfections section was modified from 1 μg/mL to 10 μg/mL in the revised manuscript. (line 398)
Comments 16: All experiments in Figure 4 need to be redone and analysed properly and the authors should present the values of the data and not the mean value.
Response 16: Thank you for your detailed review. We were redone and analyzed qPCR with all reference samples. We indicated the value of data based on the untreated control to compare the difference of mRNA expression. (Figure 4)
Comments 17: Figure 5: in Fig5B and Fig5E we need to see additional graphs with the normalised values, pNFkb/NFkb and pIkBa/IkBa, respectively.
Response 17: Thank you for your detailed review. We agree with your comment. As fer your suggestion, we added the relative protein expression of p-NF-κB/NF-κB and p-IκB-α/I-κB-α in the revised manuscript. (Figure 5B,5E)
Comments 18: Fig5C: the figure is missing data. The authors should show TLR3 levels in the absence of Poly I:C and presence of TLR3 siRNA and in the presence of Poly I:C and absence of TLR3 siRNA
Response 18: Thank you for your detailed review. We agreed with your comment. As fer your suggestion, we conducted the experiment again and replaced the new data in the revised version. (Figure 5C)
Comments 19: Fig5D: TRIF siRNA does not reduce TRIF levels at steady state. How this can be explained?
Response 19: Thank you for your comment. We agreed with you. We conducted the experiment again and replaced the data with more clearly reduced TRIF levels. (Figure 5D)
Comments 20: Figure 6: L231-233: the statement is wrong Figure 6A simply shows the levels of MAVS upon TLR3 stimulation, does not imply a link between MAVS levels and mtROS. Please delete the sentence
Response 20: Thank you for your suggestion. We agree with your comment. As you mentioned, we deleted the sentence “These results were consistent with previous data showing that poly(I:C) induces mtROS generation after 6 h, suggesting that MAVS expression may be affected by mtROS.” in the revised version.
Comments 21: Figure 6B: for how long the cells were stimulated? Controls are missing: since the Mito-TEMPO pretreatment was done for 1h, and is dissolved in DMSO, we need to see the level of MAVS after the pretreatment and before the stimulation.
Materials and Methods
L343-344: which reagents are dissolved in DMSO and what is the final concentration of DMSO. All these experiments need to show data with DMSO alone.
Response 21: I apologize again for our mistake. We didn’t use DMSO with Mito-TEMPO. We freshly prepared the poly(I:C) with sterile endotoxin-free physiological water (0.9% NaCl) provided just prior to each experiment. Sorry for the confusion. We have modified Material & Method of 4.1. Cell culture and Poly I:C treatment section in the revised version. (lines 379-381)
Comments 22: L361: siRNA experiments were done with 1ug/ml Ply I:C but in the results says 10. L365: the sentence does not make sense. Please rewrite.
Response 22: Thank you for your detailed review. We apologize for our mistake. As per your suggestion, we corrected the concentration “1 μg/mL” to “10 μg/mL” in the revised version. (line 398)
Comments 23: L370-375: need to show the dilution of the antibodies used, as in L378.
Response 23: Thank you for your suggestion. As you mentioned, we added the dilution information of the antibodies in the revised manuscript. (lines 408-414)
Comments 24: L392: ‘recruited’ please delete. The cells were transferred to tubes.
Response 24: Thanks for this precious advice. We agree with your comment. As per your suggestion, we modified “recruited” to “transferred” in the revised version. (line 431)
Comments 25: L393: The cells were centrifuged. In what solution they were resuspended for the flow cytometry analysis??
Response 25: Thank you for your comment. As per your suggestion, we added the information about solution used for resuspension in the Material & method 4.4. Flow cytometry section. The sample were resuspended using DPBS and analyzed by flow cytometry. (lines 432)
Comments 26: L400: what reagents were dissolved in medium?
Response 26: Thank you for your comment. We carefully confirm the experimental process again and we decided to delete this sentence because it was considered unnecessary. Instead, as you mentioned at Comments 27, we added detailed experiment procedure of MitoSOX in the Material and method 4.5 Confocal fluorescence microscopy section in the revised version. (lines 441-443)
Comments 27: Confocal microscopy: the authors do not describe the MitoSOX method. When it was added? We need to see the detailed experimental procedure
Response 27: Thank you for your detailed review. As for your suggestion, we added the detailed experiment procedure of MitoSOX in the Material and method 4.5 Confocal fluorescence microscopy section. (lines 441-443)
Comments 28: According to the manufacturers the MitoSOX final concentration has to be optimised for different experiments. This is crucial to maximise signal-to-noise ratio and minimise cellular toxicity. Are these experiments done? If yes, it will be nice to see them in supplementary material.
Response 28: Thank you for your valuable comment. We already optimized the concentration of mitoSOX for BEAS-2B cells. We attached the data in Supplement Figure 1. (lines 125-126, Figure S1)

Round 2
Reviewer 2 Report
Comments and Suggestions for Authors
The Authors in a relevant manner addressed my major points as well as introduced brand new figure 6 with assessing the levels of TBKs/IRF3s forms.
Although paper still has some limitations, I am supportive of the publication in current form, after carefully conducted proofreading is done - there are still some typos/minor grammatical errors.
Author Response
Comments 1: The Authors in a relevant manner addressed my major points as well as introduced brand new figure 6 with assessing the levels of TBKs/IRF3s forms.
Response: We would like to thank the reviewer for the valuable review of our manuscript.
Although paper still has some limitations, I am supportive of the publication in current form, after carefully conducted proofreading is done - there are still some typos/minor grammatical errors.
Response: Thank you for your comment. As per your suggestion, the English structure and grammar of the manuscript has been reviewed through a specialized English editing office for proof reading.

Reviewer 3 Report
Comments and Suggestions for Authors
Comments 9: Why the amount of ROS at time 0 is very high? Which is contrast to similar time point in Fig2B (confocal data)
Response 9: We measured the amount of ROS at the different concentration of poly(I:C) in Fig. 2B. The amount of ROS at time 0 was detected by FACS in Figure 2A and the shift was basic fluorescence expression of mitoSOX in BEAS-2B cells. We optimized the final concentration of MitoSOX and added this data in Supplemental Figure 1 in the revised version.
Τhe Sup.Fig.1 has no figure legend indicating how the experiment was done, meaning whic doses were tested what we see in the video. It is inconclusive.
Comments 10: Figure 3C: TNF is synthesised as pre-TNF which is cleaved to mature TNF. The antibody used in this study can detect both forms. Therefore, the authors need to present the levels of both pre-TNF and mature TNF in western blot. This will show whether treatment the synthesis or the processing to mature or both.
Response 10: Thank you for your detailed review. We agree with your comment. Based on your comment, we indicated the molecular weight in the Figure 3C. Additionally, the original image for blot we attached shows the levels of both pre-TNF-α and mature TNF-α at each molecular weight.
The uploaded western blots look like background noise. Also there are no molecular weight markers. Not accepted as reliable result
Comments 11: The western blot is not convincing that the Miso-TEMPO reduces poly I:C-induced TNF levels. There is no quantification / statistics.
Response 11: Thank you for your comment. As per your suggestion, we added the quantification data in Figure 3D.
Similar to previous. Based on the uploaded western blots look like background noise. Also there are no molecular weight markers. Not accepted as reliable result
Comments 14: Consequently the analysis DDCt method is wrong because the reference sample is not the untreated for all conditions. The untreated can be used as reference only for the Poly I:C treatment.
Response 14: Thank you for your comment. As you mentioned, we added the untreated for all conditions (TLR3 siRNA alone and Mito-TEMPO alone) and obtained the relative expression data. We added new data in the revised manuscript. (Figure 4)
We need to see the raw data, the Ct values of these experiments. Either in a table in supplementary files or on the graphs as individual points.
Comments 16: All experiments in Figure 4 need to be redone and analysed properly and the authors should present the values of the data and not the mean value.
Response 16: Thank you for your detailed review. We were redone and analyzed qPCR with all reference samples. We indicated the value of data based on the untreated control to compare the difference of mRNA expression. (Figure 4)
Same as previous comment, we need to see the raw data, the Ct values of these experiments, either in a supplementary file and on the graph
Comments 19: Fig5D: TRIF siRNA does not reduce TRIF levels at steady state. How this can be explained?
Response 19: Thank you for your comment. We agreed with you. We conducted the experiment again and replaced the data with more clearly reduced TRIF levels. (Figure 5D)
Comments 20: Figure 6: L231-233: the statement is wrong Figure 6A simply shows the levels of MAVS upon TLR3 stimulation, does not imply a link between MAVS levels and mtROS. Please delete the sentence
Response 20: Thank you for your suggestion. We agree with your comment. As you mentioned, we deleted the sentence “These results were consistent with previous data showing that poly(I:C) induces mtROS generation after 6 h, suggesting that MAVS expression may be affected by mtROS.” in the revised version.
Comments 28: According to the manufacturers the MitoSOX final concentration has to be optimised for different experiments. This is crucial to maximise signal-to-noise ratio and minimise cellular toxicity. Are these experiments done? If yes, it will be nice to see them in supplementary material.
Response 28: Thank you for your valuable comment. We already optimized the concentration of mitoSOX for BEAS-2B cells. We attached the data in Supplement Figure 1. (lines 125-126, Figure S1)
Supplentary Figure 1 has 2 videos and no figure legend.
Additionally, most western blots show no molecular weight markers and most of them have multiple bands, so they are inconclusive.
Author Response
Comments 9: Why the amount of ROS at time 0 is very high? Which is contrast to similar time point in Fig2B (confocal data)
Response 9: We measured the amount of ROS at the different concentration of poly(I:C) in Fig. 2B. The amount of ROS at time 0 was detected by FACS in Figure 2A and the shift was basic fluorescence expression of mitoSOX in BEAS-2B cells. We optimized the final concentration of MitoSOX and added this data in Supplemental Figure 1 in the revised version.
Τhe Sup.Fig.1 has no figure legend indicating how the experiment was done, meaning whic doses were tested what we see in the video. It is inconclusive.
Response: Thank you for your suggestion. According to the reviewer's comment, we added the more detailed information in the legend of the Supplemental Figure 1.
Comments 10: Figure 3C: TNF is synthesised as pre-TNF which is cleaved to mature TNF. The antibody used in this study can detect both forms. Therefore, the authors need to present the levels of both pre-TNF and mature TNF in western blot. This will show whether treatment the synthesis or the processing to mature or both.
Response 10: Thank you for your detailed review. We agree with your comment. Based on your comment, we indicated the molecular weight in the Figure 3C. Additionally, the original image for blot we attached shows the levels of both pre-TNF-α and mature TNF-α at each molecular weight.
The uploaded western blots look like background noise. Also there are no molecular weight markers. Not accepted as reliable result
Response: We agree with this comment. Therefore, we attached new data in the original image and revised the Figure 3C in the revised version. And we also added the molecular weight marker you mentioned in the data.
Comments 11: The western blot is not convincing that the Miso-TEMPO reduces poly I:C-induced TNF levels. There is no quantification / statistics.
Response 11: Thank you for your comment. As per your suggestion, we added the quantification data in Figure 3D.
Similar to previous. Based on the uploaded western blots look like background noise. Also there are no molecular weight markers. Not accepted as reliable result
Response: As per your suggestion, we revised the Figure 3C in the revised manuscript. And we also added the molecular weight marker in the revised version.
Comments 14: Consequently the analysis DDCt method is wrong because the reference sample is not the untreated for all conditions. The untreated can be used as reference only for the Poly I:C treatment.
Response 14: Thank you for your comment. As you mentioned, we added the untreated for all conditions (TLR3 siRNA alone and Mito-TEMPO alone) and obtained the relative expression data. We added new data in the revised manuscript. (Figure 4)
We need to see the raw data, the Ct values of these experiments. Either in a table in supplementary files or on the graphs as individual points.
Response: Thank you for your comment. As per your opinion, we added the Ct value in supplementary files.
Comments 16: All experiments in Figure 4 need to be redone and analysed properly and the authors should present the values of the data and not the mean value.
Response 16: Thank you for your detailed review. We were redone and analyzed qPCR with all reference samples. We indicated the value of data based on the untreated control to compare the difference of mRNA expression. (Figure 4)
Same as previous comment, we need to see the raw data, the Ct values of these experiments, either in a supplementary file and on the graph
Response: Thank you for your comment. As per your suggestion, we added the Ct value in supplementary files.
Comments 28: According to the manufacturers the MitoSOX final concentration has to be optimised for different experiments. This is crucial to maximise signal-to-noise ratio and minimise cellular toxicity. Are these experiments done? If yes, it will be nice to see them in supplementary material.
Response 28: Thank you for your valuable comment. We already optimized the concentration of mitoSOX for BEAS-2B cells. We attached the data in Supplement Figure 1. (lines 125-126, Figure S1)
Supplentary Figure 1 has 2 videos and no figure legend.
Response: Based on your comments, Supplement Figure 1 of the newly revised supplement file contains data that optimizes MitoSOX final concentration. Therefore, we moved the video file to Supplementary Figure 2.
Additionally, most western blots show no molecular weight markers and most of them have multiple bands, so they are inconclusive.
As per your suggestion, we added the molecular weight marker in the revised version.

Round 3
Reviewer 3 Report
Comments and Suggestions for AuthorsΤhank you for the response.